# Tribological Investigation of the Effect of Nanosized Transition Metal Oxides on a Base Oil Containing Overbased Calcium Sulfonate

Álmos Dávid Tóth [1], Hajnalka Hargitai [2] and Ádám István Szabó [1,*]

1 Department Propulsion Technology, Széchenyi István University, Egyetem tér 1,
H-9026 Győr, Hungary; toth.almos@sze.hu
2 Department of Materials Science and Technology, Széchenyi István University, Egyetem tér 1,
H-9026 Győr, Hungary; hargitai@sze.hu
* Correspondence: szabo.adam@sze.hu

**Abstract:** In this study, copper(II) oxide, titanium dioxide and yttrium(III) oxide nanoparticles were added to Group III-type base oil formulated with overbased calcium sulfonate. The nanosized oxides were treated with ethyl oleate surface modification. The tribological properties of the homogenized oil samples were tested on a linear oscillating tribometer. Friction was continuously monitored during the tribological tests. A surface analysis was performed on the worn samples: the amount of wear was determined using a digital optical and confocal microscope. The type of wear was examined with a scanning electron microscope, while the additives adhered to the surface were examined with energy-dispersive X-ray spectroscopy. From the results of the measurements, it can be concluded that the surface-modified nanoparticles worked well with the overbased calcium sulfonate and significantly reduced both wear and friction. In the present tribology system, the optimal concentration of all three oxide ceramic nanoadditives is 0.4 wt%. By using oxide nanoparticles, friction can be reduced by up to 15% and the wear volume by up to 77%. Overbased calcium sulfonate and oxide ceramic nanoparticles together form a lower friction anti-wear boundary layer on the worn surfaces. The results of the tests represent another step toward the applicability of these nanoparticles in commercial engine lubricants. It is advisable to further investigate the possibility of formulating nanoparticles into the oil.

**Keywords:** lubricant; tribology; nanoparticle; detergent; cupric oxide; titanium dioxide; yttrium oxide; engine oil

## 1. Introduction

The requirements for modern internal combustion engines not only indicate the development of engines but also the development of lubricants. During the natural use of engine oils, a small amount is sufficient, while its additives produce pollutants whose emissions must be regulated. Studies have summarized the positive tribological properties of oxide ceramic nanoparticles, and their possible application as engine lubricant additives [1]. This aroused the interest of researchers to investigate the possibility of using nanoparticles—which promise significant potential—instead of traditional engine oil additives [2]. A review by Akl et al. summarized the articles that examine the results of research supporting the use of different nanoparticles as engine lubricating oil additives from different aspects [3]. The friction performance exerted by a nanoparticle depends not only on the material of the nanoparticle [4] but also on its morphological shape [5]. This article presents a result that is part of this engine oil development process, during which the compatibility of the overbased calcium sulfonate detergent additive and three different transition metal oxide (copper(II) oxide, titanium dioxide and yttrium(III) oxide) nanoparticles in base oil was investigated by using experimental tribotests.

Detergents are a group of engine lubricant additives whose tasks are to prevent the formation of deposits, keep oil-insoluble particles suspended and neutralize acidic components (these would degrade the lubricant and increase corrosion wear). Overbased calcium sulfonate (OBCS) is a common anti-wear detergent additive in modern motor oils [6]. OBCS generates a protective calcium oxide tribofilm on the contact surfaces [7], this boundary layer provides the base body with favorable friction-reducing and anti-wear properties. Richardson et al. demonstrated—with the help of real, rolling tribological tests—that the amount of fatigue wear of bearing steel can be reduced by using OBCS [8]. Zhongyi et al. reported the possibility of using OBCS as an extreme pressure additive. During their research, they proved that a chemically complex mixed reaction boundary film is formed on the contact surfaces [9]. During the AFM examination of the boundary film, it was proven that it forms only on the higher parts of the surface, not in the surface roughness valleys [10]. Additional tribochemical and layer-formation studies were performed with PM-IRRAS spectroscopy [11,12] and a ToF-SIMS analysis [13,14]. The specific effects and properties of OBCS occurring in the vehicle industry were also investigated as its interaction with ZDDP in PAO base lubricant [15,16], its influence of water on the tribological properties in transmission fluids [17] and as an additive in complex greases with nanoparticles [18,19].

There have been many studies on the effects and roles of copper(II) oxide in the tribological system in the case of different bases. Alves et al. found that the CuO nanoparticles are incorporated from the PAO base lubricant into the tribofilm with tribosintering. The formed boundary layer resulted in a lower friction coefficient and higher wear resistance. The surface morphology of the CuO-containing tribofilm changes depending on the nanoparticle concentration used; based on their results, a CuO content of 0.1 wt% is considered optimal [20]. The authors investigated the tribological properties of CuO nanoparticles in another common automotive base oil—Group III-type oil. It was found that the CuO nanoparticles achieve the friction-reducing effect (~15% reduction) with the generation of elemental copper produced during the triboreduction of CuO. The optimal CuO concentration was found to be 0.5 wt%, and the wear volume was reduced by 69% [21]. The friction- and wear-reducing effect of nanosized CuO has also been demonstrated in liquid paraffin [22] and formulated lubricating oils [23,24]: in the case of 10W-30 lubricant doped with nanosized CuO, an EDX analysis showed that an elemental copper layer forms on the surfaces [25]. Jatti et al. investigate the effect of many different factors on the reduction in the coefficient of friction. In addition to the application of different CuO nanoparticle concentrations, the amount of friction was investigated when different loads and different relative velocities were applied [23]. For long-term sustainability, many studies are going on in the field of developing biolubricants. CuO nanoparticles prove to be a tribologically effective additive in plant-based lubricating oils such as palm oil [26], brassica oil [27] or sal oil [28], but also in water-based lubricants [29]. Wu et al. experimentally proved that CuO cooperates well with the thickening agent in greases, and although the effect depends on the composition of the grease, it creates a fine and smooth worn surface overall [30]. Ta et al. reported that the lubrication mechanisms of CuO when used as nanoadditives in ionic liquid are tribo-sintering and a third body with pure rolling mechanisms, respectively [31].

Titanium dioxide has already been proven in numerous studies to have a positive effect on reducing friction and wear; therefore, it has strong potential in the area of formulated engine oil development [32,33]. Birleanu et al. mixed $TiO_2$ nanoparticles into 10W-30 motor oil. With four-ball tribometer experiments, it was determined that friction can be reduced by up to 80% by using $TiO_2$ nanoparticles as a result of its ball-bearing operation. Wear can be reduced by using $TiO_2$ nanoparticles because it helps to form a smooth tribofilm [34]. Using $TiO_2$ nanoparticles added to 20W-40-type engine oil, Shaik et al. showed, on a pin-on-disc tribometer, that titania nanoparticles reduce shear resistance, so friction can be reduced by up to 86.1% when 0.4 wt% nanoparticles are used [35]. Further automotive industry research has also positively confirmed the possibility of using titania nanoparticles as an additive on cylinder walls using 5W-30 commercial motor oil [36], polyalphaolefin-based gel lubricant [37], lithium grease [38], liquid paraffin [39,40] and SAE30 fully formulated

lubricating oil [41]. Based on the unanimous positive results of the widely performed tribological tests of titania nanoparticles, they belong to the additives recommended for engine lubrication formulation tests.

Nanoscale yttrium(III) oxide is not being researched as a possible engine lubricant additive. In previous works, the authors reported on the excellent anti-wear effect of nanosized $Y_2O_3$: when used in Group III-type base oil at a concentration of 0.5 wt%, a 45% reduction in wear scar diameter and a 90% reduction in wear volume were achieved. The tribosintered, yttrium-containing boundary layer formed on the worn surface was responsible for the wear-reducing effect [42]. Due to the high cost of $Y_2O_3$, the authors replaced part of it with zirconium dioxide. The synergistic effect between $Y_2O_3$ and $ZrO_2$ was demonstrated when, together, the two oxide ceramic mixtures also resulted in a 90% reduction in wear [43]. Based on the results so far, it is worthwhile to continue researching $Y_2O_3$ nanoparticles for lubricating oil formulation, because they promise a very strong wear-reducing effect.

The main aim of this paper is to investigate the tribological effect of three selected nanoparticles (CuO, $TiO_2$ and $Y_2O_3$) in overbased calcium sulfonate (OBCS) containing base oil. The main novelty of this paper is that there is no available research article about the correlation between conventional engine oil additives and nanoparticles. The researchers usually homogenized their nanoparticles into neat base oil or a fully formulated lubricant, but they did not investigate how the nanoparticles work in the presence of the conventional engine lubricant additives separately.

## 2. Materials and Methods

All of the transition metal oxide nanoceramics used for the tests had a particle size below 100 nm in diameter. The copper(II) oxide nanoparticles came in the form of APS powder from Thermo Fisher (Kandel, Germany) GmbH; the purity of the nanopowder was >99% and the particle size was between 30 and 50 nm. The titanium dioxide nanoparticles used in the measurements came in APS powder form from Alfa Aesar GmbH, Karlsruhe, Germany. The $TiO_2$ nanopowder was spherical in shape, had an anatase crystal structure, was >99.9% pure and had a grain size of 32 nm. The used yttrium(III) oxide nanoparticles came in APS powder form from Reanal Laborvegyszer Kereskedelmi Kft., Budapest, Hungary. The $Y_2O_3$ nanopowder had a spherical shape, body-centered cubic crystal structure, purity of >99.999% and particle size of 50 nm. Figure 1 shows SEM images of the bulk powder of the nanoparticles under a magnification of 1000 (CuO nanopowder on the left, $TiO_2$ nanopowder in the middle and $Y_2O_3$ nanopowder on the right). The nanoparticles presented in this article were selected according to their purchasability and the previous studies of the authors with the same methodology.

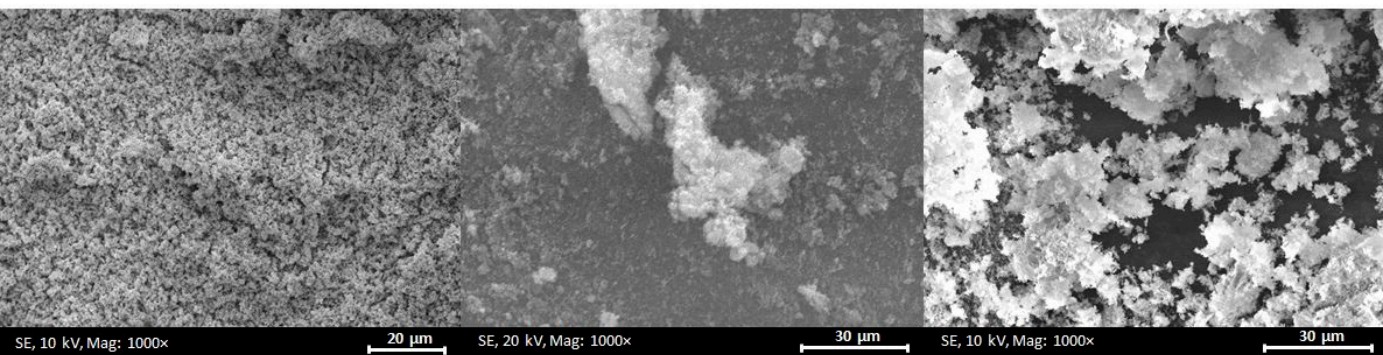

**Figure 1.** SEM images of the bulk powder of the applied oxide nanoparticles under a magnification of 1000: CuO nanopowder on the left, $TiO_2$ nanopowder in the middle and $Y_2O_3$ nanopowder on the right.

The nanolubricant used for the measurements was based on a Group III-type base oil, in which the manufacturer (MOL-LUB Kft., Almásfüzítő, Hungary) formulated 1 wt% overbased calcium sulfonate. The modern engine oils with the SAE viscosity classification of 0W-20 contain this amount of overbased calcium sulfonate additive, based on the received information for the Hungarian oil manufacturer MOL. Group III-type base oil is a lubricant produced using hydrocracking, which is a mixture of alkanes with a carbon chain length of C20–C50. The overbased calcium sulfonate (OBCS) detergent used is a common additive in engine lubricants; its task is to help keep engines clean and prolong the lifetime of the fluid. OBCS offers detergency, antioxidant performance and friction modification, providing a source of basicity to formulations. OBCS consists of an inner $CaCO_3$ core and a surface-modifying sulfonate shell. The basicity of partially formulated lubricating oil is determined with the amount of overbased detergent, which is expressed in the total base number (TBN). The TBN value of the oil sample used during the measurements was 400 [44].

Oxide ceramic nanoparticles can easily agglomerate in the oil, which can lead to their sedimentation, thereby making the oil sample inhomogeneous, and the concentration at the point of use may change. To avoid this phenomenon, the nanoparticles need to be surface modified, so that the oil sample can maintain its homogeneity over time. During the modification of the surface of the nanoparticles, a coating is created on their passive surface, which makes it easier to mix in oil and prevents the coagulation and flocculation (agglomeration) of the individual nanoparticles. During the present research, the oxide ceramic nanoparticles were treated with ethyl oleate surface modification. During the ethyl oleate surface modification, 90% pure ethyl oleate and 99.6% pure ethyl alcohol were added to the APS nanopowder. This mixture was kept at a temperature of $75 \pm 5$ °C with continuous stirring for 2 h. At this temperature, during the esterification reaction of ethyl alcohol and oleic acid, ethyl oleate was formed, which coats the surface of the nanoparticles. The already surface-modified nanopowder was dried at 80 °C. The dried nanopowder was homogenized with toluene into the Group III base oil containing 1 wt% OBCS. With continuous stirring, toluene evaporated from the lubricant sample in 16 h. After this, the final homogenization of the oil sample was carried out with an ultrasonic mixer for 15 min at a temperature of 50 °C. At the end of the ultrasonic mixing, the oil sample was homogeneous, stable, ready for tribometer tests and under continuous magnetic stirring until the start of the measurement. The authors described the process in more detail in a previous article and proved that the oils made with surface-modified nanoadditives using this method are homogeneous, form a stable dispersion and are suitable for tribometer tests [45].

For the experimental analysis of nanoparticle-reinforced lubricant samples, the ball-on-disc module of an Optimol SRV®5 tribometer (Optimol Instruments GmbH., Munich, Germany) was used, which was placed in the Tribological Laboratory of Széchenyi István University in Győr, Hungary. This equipment enabled the realization of the tribological loads and relative movement of the tribosystem with a high precision and this tribometer can be used both for scientific and development purposes. To ensure the highest repeatability and reproducibility of the research, standardized ball and disc specimens were used, which correlates with ISO 19291:2016 [46]. The Ø10 mm ball specimens were made of 100Cr6 material, their surface was polished for Ra 0.025 μm and their hardness was 61 HRC. Besides this, the Ø24 mm × 7.9 mm disc specimens were also manufactured from 100Cr6 material, but the specimens were also vacuum arc melted. The flat contacting surface of the discs was ground and lapped for Ra 0.035 μm and their hardness was 62 HRC. Each specimen was thoroughly cleaned in an ultrasonic cleaner device and brake disc cleaner medium at a 50 °C temperature for 15 min to remove the unnecessary contaminations from the surface and provide equal initial conditions before placing them into the measurement chamber of the tribometer. The homogenized nanolubricant was filled into the continuous oil circuit of the tribometer. A self-developed tribological testing method [47] was used, which is based on the ISO 19291:2016 standard but some modification was executed (adding a continuous oil circuit) to provide more engine-relevant conditions for the testing. The tribological investigation is two-phase testing with the necessary preheating phase. During

preheating, the specimens were pushed together with the normal force of 50 N and both specimens and oil were heated up to a 100 °C temperature during the realized oil flow rate of 225 mL/h. The main purpose of this preheating phase is to provide equal conditions for every individual measurement and specimen to ensure the highest reproducibility and repeatability. The first testing phase is a short low-load phase where a 50 N normal force is applied on the surfaces next to the realized sinusoidal oscillation movement (a 1 mm stroke and 50 Hz frequency) with the previously set 100 °C temperature values. The aim of this step is to prepare the necessary lubricating film between the contacting surfaces before higher normal forces are applied. The last step is a long-lasting (2 h) test with a higher loading force (100 N). During phases one and two, a continuous data saving mode was used in the case of every single setup and measured parameters with the data saving frequency of 1 Hz. However, the coefficient of friction values was recorded every 40 µs to provide high-speed saving data for calculating the friction absolute integral value (FAI), which is the absolute integral value of the measured friction coefficient data calculated with the following formula:

$$\text{FAI} = \frac{1}{s_{max}} \cdot \int_{s_0}^{s_{max}} |\mu(s)| ds \tag{1}$$

where 's' is the applied stroke and 'µ' is the high-speed measured friction coefficient data.

The tribological experiments were always followed by a thorough microscopical analysis of the loaded surfaces to gather information about the wear processes and mechanisms that happened during the tribological tests. The microscopical investigation was carried out in the Surface Analytic Laboratory and the Material Testing Laboratory of Széchenyi István University in Győr, Hungary. Before the microscopical investigation, the specimens were thoroughly cleaned in an ultrasonic cleaner to remove oil contamination and wear debris. The cleaning process was carried out in a 50 °C heated brake disc cleaner with a time duration of 15 min. The worn surfaces on both ball and disc specimens were firstly documented with a Keyence VHX-1000-type digital microscope (Keyence International, Mechlin, Belgium), which proved the measuring option of the mean wear scar diameter (MWSD) described in the ISO 19291:2016 standard: the wear scar diameter was measured on the ball specimens parallel and perpendicular to the sliding direction and the average of these two values equals the mean wear scar diameter in the micrometer unit. The worn surfaces on the disc specimens were 3D scanned with a high-precision confocal microscope (Leica DCM 3D, Leica Camera AG, Wetzlar, Germany) to calculate the missing volume: the worn surface was enclosed to provide the borderline between the worn and not worn surface area, a flat theoretical surface was placed on the not worn surface points of the surface using the least squares method and the volume under this theoretical plane inside the enclosed area was calculated. Furthermore, a scanning electron microscope (Hitachi S-3400N, Hitachi Ltd., Chiyoda, Tokyo, Japan) was used to prepare high-magnification images of the worn surfaces to investigate the wear mechanisms that happened during the tribotest and its Bruker EDX (energy dispersive X-ray spectroscopy, Billerica, MA, USA) sensor was also used to identify the distribution of the nanoceramic particles inside the worn surfaces. The scanning electron micrographs were taken at two characteristic points of the wear mark on the disc: in the middle point of the wear track (highest relative speed) and the dead center (zero relative speed). All SEM and EDX images were taken at a magnification of 1000 using an accelerating voltage of 20 kV and a working distance of 10.4 mm.

Four independent experiments were carried out with every lubricant sample variation to provide the necessary measurement results to calculate statistical values: both AVERAGE and 'STDEV.P' data were calculated from the raw data with the help of Excel functions.

## 3. Results

### 3.1. Cupric Oxide

During the analysis of the measured friction coefficient values (Figure 2), it can be defined that the investigated cupric oxide can reduce the frictional losses of the used ball-on-disc tribosystem. A clear tendency can be defined: the addition of cupric oxide into

the oil samples reduced the frictional losses until a 0.4 wt% concentration and, starting at 0.5 wt%, a friction increase is measurable. A similar tendency can be observed in the case of the defined mean wear scar diameter values as well (Figure 3): a continuous decrease until 0.4 wt% and an increase at 0.5 wt%. The optimum concentration can be defined at 0.4 wt% because both the friction coefficient and mean wear scar diameter values were the lowest at this concentration. The oil sample with the optimum concentration reduced the measurable friction coefficient value by 4% and the mean wear scar diameter value by 27% compared to the results of the reference sample without additional nanoparticles.

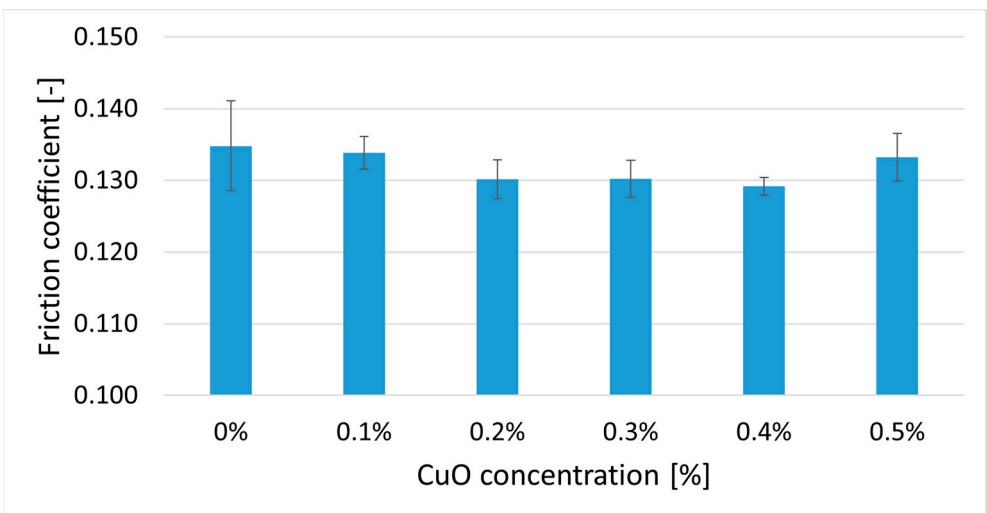

**Figure 2.** Measured friction coefficient values caused by the nanolubricants with different cupric oxide concentrations.

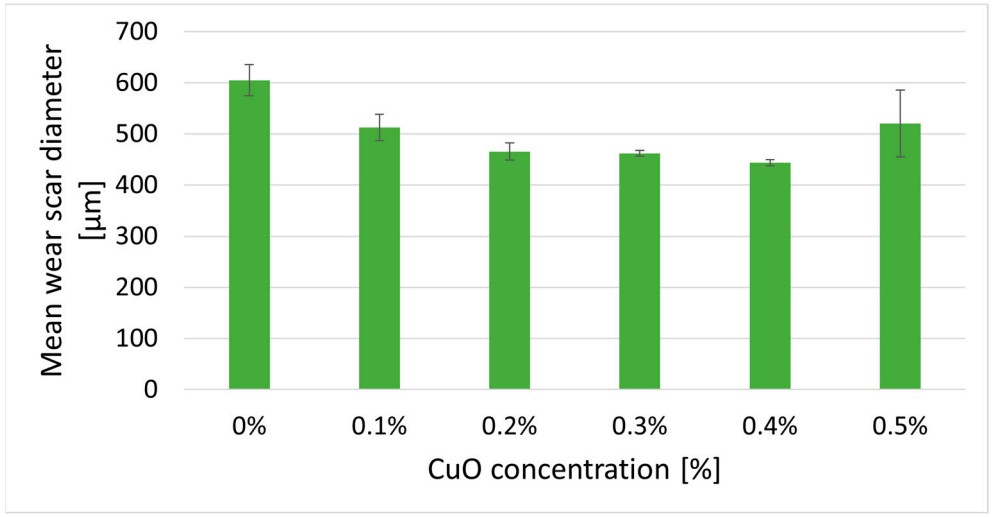

**Figure 3.** Measured mean wear scar diameter values caused by the nanolubricants with different cupric oxide concentrations.

Figure 4 illustrates the captured digital microscope images of the worn surfaces of 0.4 wt% cupric oxide added and of the reference (each of the images were captured with the same setups of the microscope). During the comparison of these images, the significant difference in the diameter of the rounded wear area on the ball specimens and in the width of the wear scar on the disc specimen is visible. Furthermore, some original surface grooves are also seeable in the image with about 0.4 wt% variation while almost the complete original surface was removed in the case of the reference oil sample. No critical damage or

wear can be observed in this image; the main wear mechanism seems to be abrasion parallel to the sliding direction. This information confirms the previously mentioned positive anti-wear properties of this cupric oxide nanoparticle. The characteristic yellowish color of the elemental copper produced during the triboreduction of CuO during wear can be observed on the ball test specimen [21].

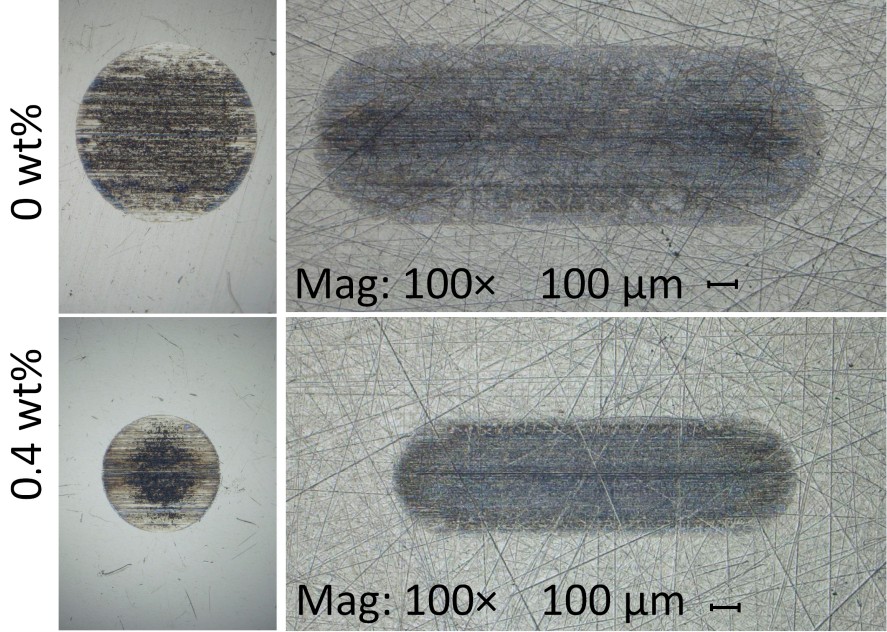

**Figure 4.** Digital microscope images of the worn surfaces of the ball (**left**) and disc (**right**) specimens with a 0 (**top**) and 0.4 wt% (**bottom**) cupric oxide concentration nanolubricant sample.

The analysis of the missing volume on the disc specimens proved the accuracy of the investigated cupric oxide nanoparticles. As can be observed in Figure 5, the lowest wear volume was defined in the case of the oil sample with a 0.4 wt% CuO concentration with a decreasing potential of 48%. The defined wear volume was only increased in the case of one concentration: the 0.2 wt% sample caused a 50% wear volume increase, compared to the reference sample without cupric oxide.

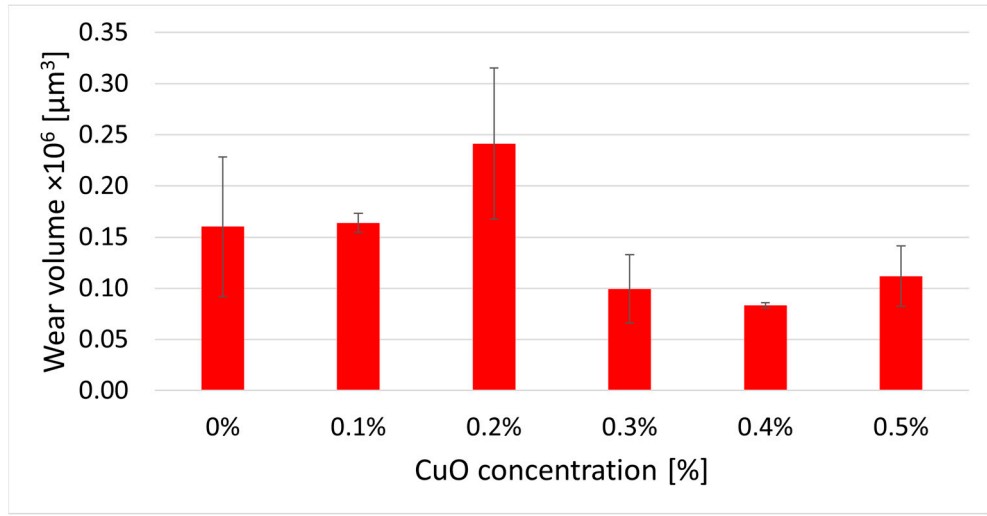

**Figure 5.** Measured wear volume values caused by the nanolubricants with different cupric oxide concentrations.

Figure 6 shows the scanning electron microscopy images of the wear track of the disc tested with a 0.4 wt% CuO nanoadditive (left column), supplemented with the Cu locations measured with EDX (right column). The figures in the upper row were taken in the middle point of the wear track, while the lower ones are in the dead center of the ball. Based on the electron microscopy images, it can be established that the wear is minor, and the original surface structure can still be noticed. Abrasion marks visible in the direction of ball movement are the main type of wear. It can be observed on the surface that there are large dark spots and slightly fewer light areas, which are mainly visible in the deeper parts. In the dark areas, overbased calcium sulfonate created the protective layer—already reported by many scientists [7,9]. This phenomenon, according to which the OBCS forms a protective boundary layer on the upper layer of the surface, is related to the results of a study by Topolovec-Miklozic [10].

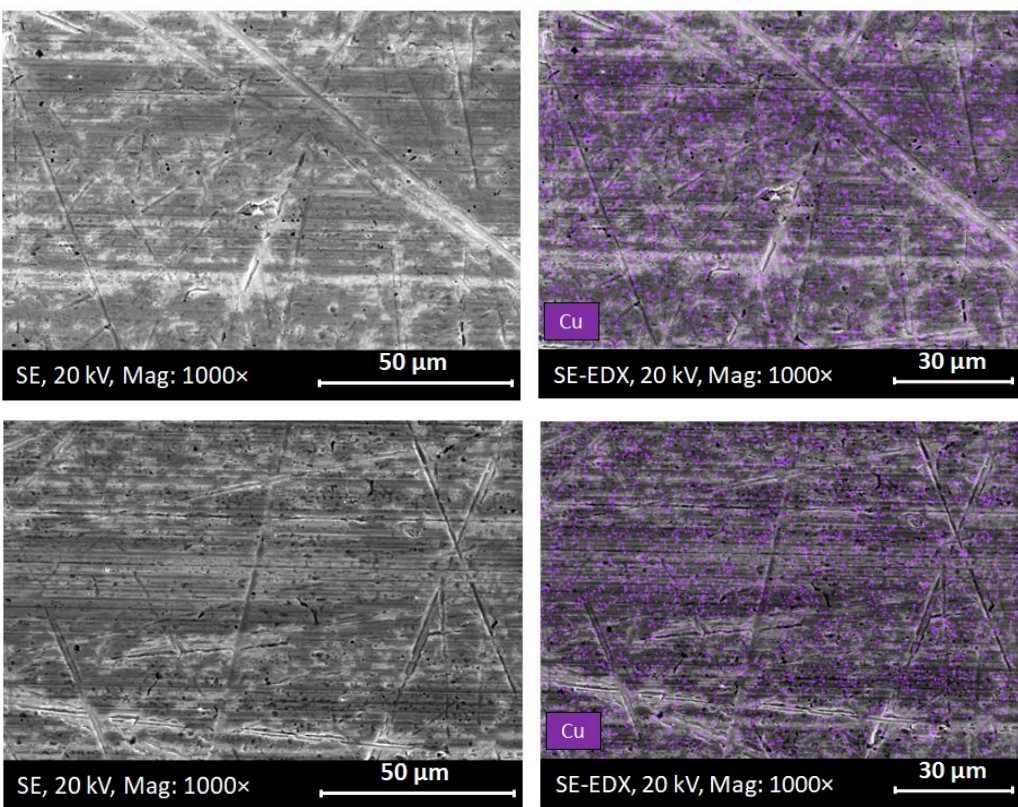

**Figure 6.** SEM and EDX images of the worn surface of the disc specimens measured with 0.4 wt% cupric-oxide-added samples from its stroke-middle section (**top**) and dead center (**bottom**) areas with the horizontal sliding direction.

With the help of energy-dispersive X-ray spectroscopy, it is possible to show the occurrence of specific elements on the map and to quantify the proportion of elements on the surface (see the quantification table in Table 1). The dark protective layer created with OBCS at the dead center, where the relative speed is 0 or close to 0, covers the surface to a greater extent (Ca = 1.11 norm.wt%) than in the middle of the wear track (Ca = 0.95 norm.wt%). Since neither the material of the test specimens nor the oil contains components containing copper, traces of CuO nanoparticles can be examined using elemental copper during the EDX analysis. Copper is integrated into the boundary layer during tribochemical reactions, which can be tribosintering or triboreduction depending on the conditions. The distribution of the copper and, with it, the nanoparticle are uniform over the entire surface, both in the images taken in the center of the dead center and the wear mark. According to the quantification, there is uniformly 0.37 norm.wt% copper on the surface in both areas.

**Table 1.** Results of the SEM-EDX analysis from the investigated two different areas on the disc specimen measured with CuO-added lubricant samples; the result numbers are presented in normalized mass percent.

| Element | Stroke-Middle Section | Dead Center |
|---------|----------------------|-------------|
| Fe | 91.51 | 90.07 |
| Cr | 1.68 | 1.69 |
| Si | 0.34 | 0.36 |
| O | 2.70 | 3.80 |
| C | 2.32 | 2.48 |
| Ca | 0.95 | 1.11 |
| S | 0.13 | 0.12 |
| Cu | 0.37 | 0.37 |

In summary, the surface-modified CuO nanoadditive with ethyl oleate works properly when used in Group III base oil with a 1 wt% OBCS content. A minor friction-reducing and a good MWSD-reducing effect can be observed when used in a concentration of 0.2–0.4 wt%. In the case of higher CuO concentrations (0.3–0.5 wt%), a good wear volume reduction effect is achieved. Based on the tribological results, the 0.4 wt% CuO concentration can be considered optimal, which resulted in a 4% lower friction, 27% lower wear diameter and 48% lower wear volume. Based on the friction, wear and EDX results, a similar correlation can be seen in the functioning of nanoparticles with the work of Alves [20].

CuO and OBCS both work well together, and both can exert their effect. CuO can be incorporated into the boundary layer, helping the tribological properties, while OBCS can create the protective CaO layer on the surface. The rate of wear is low, and its typical type is abrasive. A slight antagonistic effect is observed from OBCS towards the friction-reducing effect of CuO. In comparison with the previous work of the authors, it can be concluded that the CuO nanoadditive in the presence of OBCS is not capable of triboreduction to the same extent as without it [21]. As a result of this, the occurrence of large-scale elemental copper spots is not typical (although it can be seen in the shot of the surface of the ball), and the copper is rather uniformly sintered on the surface. As a result, it was not able to exert as much of a friction-reducing effect as without OBCS. The combined application of the two additives resulted in a boundary layer with good wear resistance properties.

### 3.2. Titanium Dioxide

The investigation results of the titanium dioxide nanoparticles revealed that these nanoparticles have similar tribological properties to the cupric oxide presented in Section 3.1. Both the friction coefficient (Figure 7) and the mean wear scar diameter (Figure 8) values are showing a decreasing tendency, especially at higher concentrations (0.4 and 0.5 wt%). The 0.2 wt% sample can be called an exception because it slightly increased both measurement data values. The optimum concentration cannot be defined so easily, because the results of 0.4 and 0.5 wt% samples are almost identical: the 0.4 wt% sample provides a slightly lower friction (−4%), while the 0.5 wt% provides a slightly lower MWSD (−27%).

Figure 9 presents the wear scar images of the 0.4 wt% titania-added lubricant sample and its comparison to the reference results. The effect of the titania-containing samples is visible: not so deep wear scars because some of the original grooves can be seen. The lighter-color areas show the presence of titania, and the dark areas show that the OBCS-generated protective boundary layer was formed on the contact surfaces during the tribotest. The main wear mechanisms are abrasion and polishing.

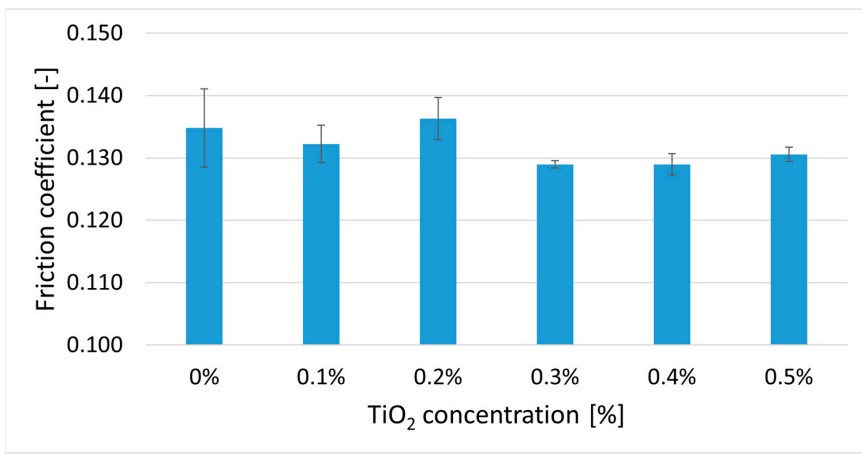

**Figure 7.** Measured friction coefficient values caused by the nanolubricants with different titanium oxide concentrations.

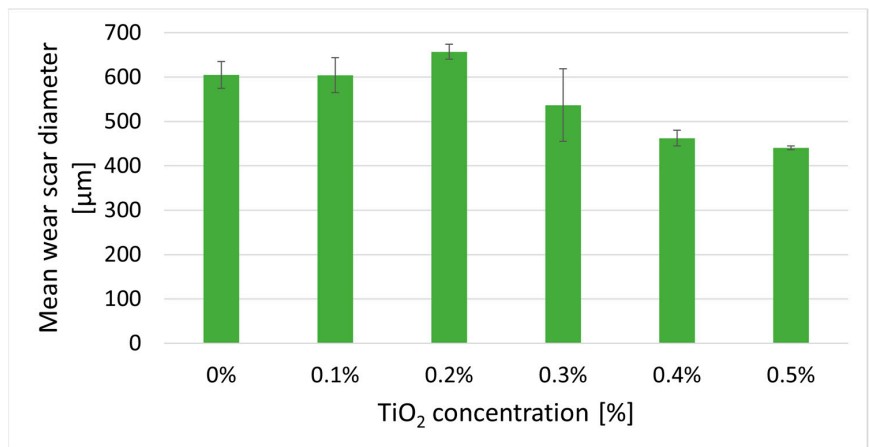

**Figure 8.** Measured mean wear scar diameter values caused by the nanolubricants with different titanium oxide concentrations.

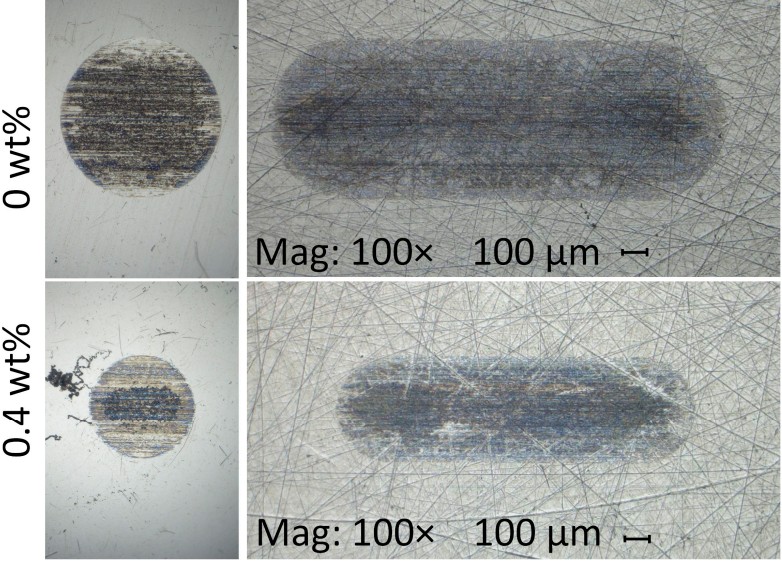

**Figure 9.** Digital microscope images of the worn surfaces of the ball (**left**) and disc (**right**) specimens with a 0 (**top**) and 0.4 wt% (**bottom**) titanium oxide concentration nanolubricant sample.

Analyzing the measured wear volume data (Figure 10), it can be stated that the higher-$TiO_2$-concentration samples provided the least missing volume from the contact surface after the executed tribotests. The optimum concentration is also quite difficult to define because both the 0.4 and 0.5 wt% concentration samples provided around 60% of wear volume reduction. The low-nanoparticle-concentration samples provided a significantly higher wear volume, and the 0.1 wt% sample caused the highest wear with a volume increase of 10% compared to the results with the reference sample without additional titania nanoparticles.

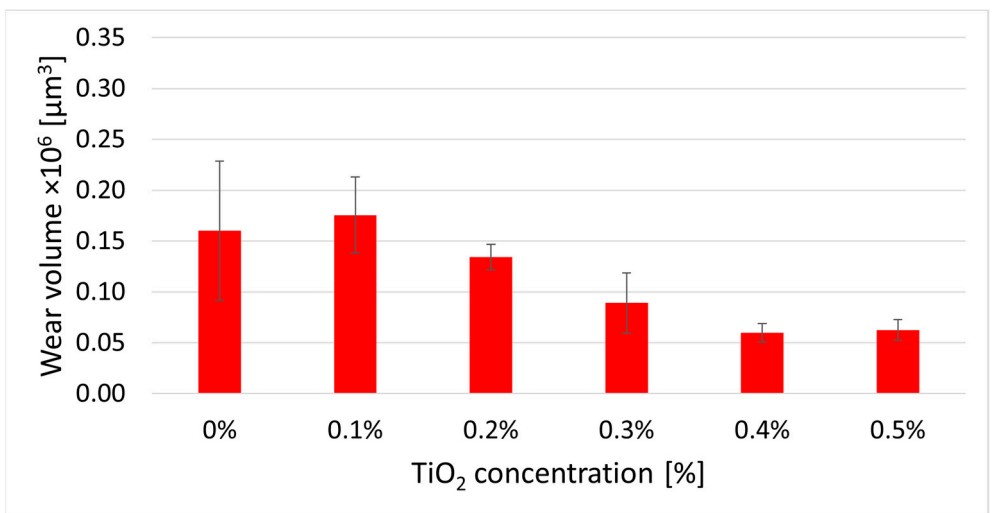

**Figure 10.** Measured wear volume values caused by the nanolubricants with different titanium oxide concentrations.

Figure 11 shows the scanning electron microscopy images of the wear track of the disc tested with a 0.4 wt% $TiO_2$ nanoadditive (left column), supplemented with the Ti locations measured with EDX (right column). The figures in the upper row were taken in the middle point of the wear track, while the lower ones are in the dead center of the ball. The results are very similar to the measurements with CuO nanoparticles. Based on the electron microscopy images, it can be established that the wear is minor, and the original surface structure can still be observed. Polishing and abrasion marks visible in the direction of ball movement are the main type of wear. It can be observed on the surface that there are large dark spots and slightly fewer light areas, which are mainly visible in the deeper parts. In the dark areas, overbased calcium sulfonate created the protective layer.

Table 2 shows the elemental quantification table of measurements performed with 0.4 wt% titania nanoadditives. Similar to what was experienced with the CuO nanoadditive, the boundary layer created with OBCS, in this case, is also larger at the dead spots (the larger proportion of dark areas can be observed in Figure 11) in the center of the wear mark Ca = 0.63 norm.wt%, while in the dead spot of the ball, Ca = 0.63 = 1.05 norm.wt% is found. Since neither the material of the test specimens nor the oil contains components containing titanium, traces of $TiO_2$ nanoparticles can be examined using elemental titanium during the EDX analysis. Based on the EDX map of titanium, it can be established that titania is found in a uniform distribution and almost the same amount (Ti = 0.41–0.42 norm.wt%) in the entire area of the wear mark. Titania is tribosintered into the boundary layer during tribochemical reactions.

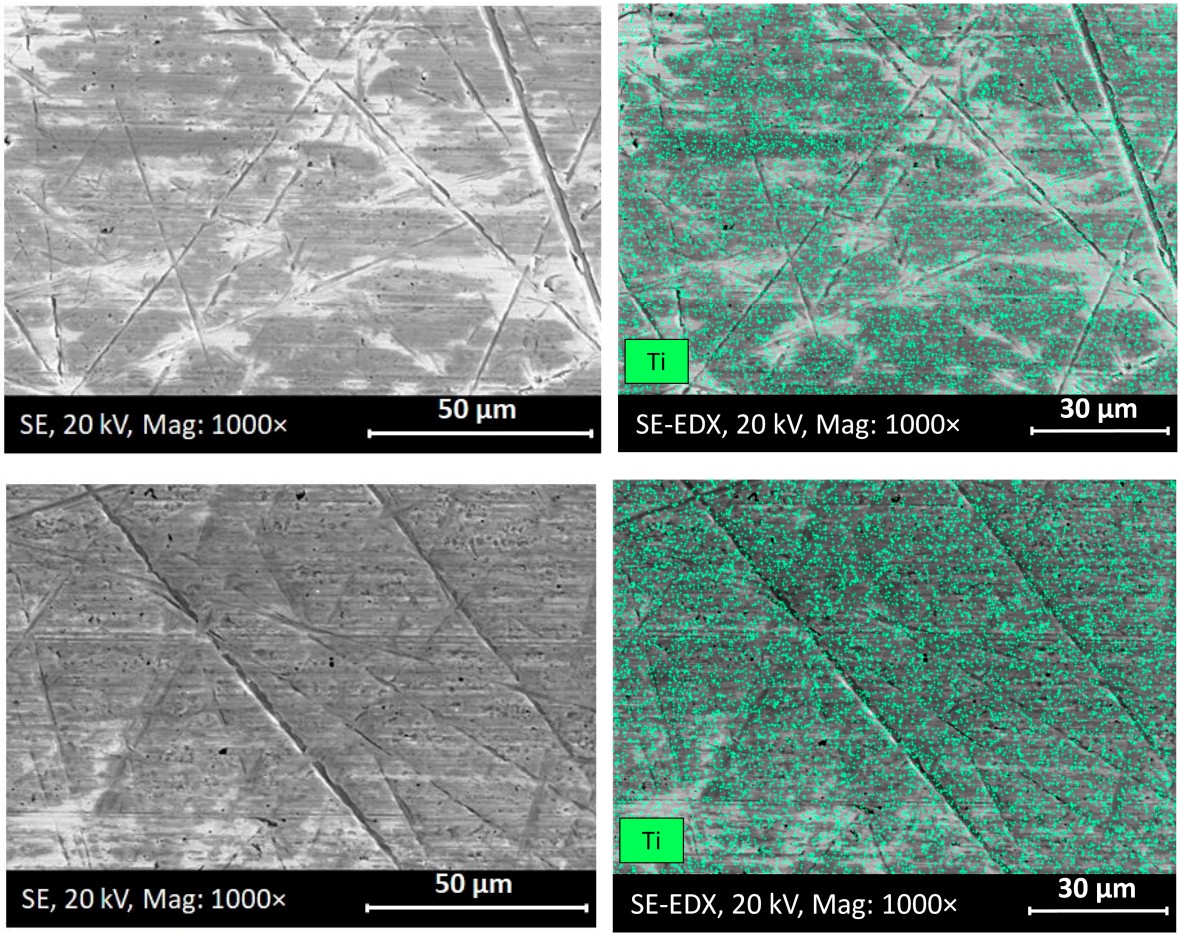

**Figure 11.** SEM and EDX images of the worn surface of the disc specimens measured with 0.4 wt% titanium-oxide-added samples from its stroke-middle section (**top**) and dead center (**bottom**) areas with the horizontal sliding direction.

**Table 2.** Results of the SEM-EDX analysis from the investigated two different areas on the disc specimen measured with $TiO_2$-added lubricant samples; the result numbers are presented in normalized mass percent.

| Element | Stroke-Middle Section | Dead Center |
|---------|----------------------|-------------|
| Fe | 91.72 | 88.12 |
| Cr | 1.73 | 1.68 |
| Si | 0.28 | 0.31 |
| O | 2.72 | 5.41 |
| C | 2.44 | 2.94 |
| Ca | 0.63 | 1.05 |
| S | 0.07 | 0.09 |
| Ti | 0.42 | 0.41 |

In summary, the surface-modified $TiO_2$ nanoadditive with ethyl oleate works properly when used in Group III base oil with a 1 wt% OBCS content. When using higher $TiO_2$ nanoparticle concentrations (0.3–0.5 wt%), better tribological effects were observed: a slight friction reduction effect, good mean wear scar diameter reduction effect and excellent wear volume reduction effect. Based on the tribological results, the 0.4 wt% $TiO_2$ concentration can be considered optimal, which resulted in a 4% lower friction, 24% lower wear diameter and 63% lower wear volume. Both titania and OBCS work well together, and both can exert their effect. Titania can be incorporated into the boundary layer, helping the tribological

properties, while OBCS can create the protective CaO layer on the surface. The rate of wear is low, and its typical type is abrasive with polishing effects. Neither an antagonistic nor a synergistic effect can be established between titania and OBCS.

### 3.3. Yttrium Oxide

Compared to the previous two subchapters presented with nanoparticles, nanoscale yttria particles show a completely different tendency in the case of the frictional (Figure 12) and wear losses (Figure 13). The measurable friction coefficient values were more drastically reduced until a 0.4 wt% concentration (a −15% friction coefficient reduction) and a dramatic raise was observed at a 0.5 wt% yttria concentration. This tendency can also be defined during the analysis of mean wear scar diameter values, where the measurable MWSD was decreased by 23% at the concentration of 0.4 wt%. The optimum concentration can be defined at 0.4 wt%, and this NP concentration sample provided the highest friction and wear decrease. It is visible that above the optimum concentration, both friction coefficient and wear losses started to raise, which can be explained with the over-satiation of nanoparticles inside the tested oil samples. The too high amount of nanoparticles can form larger agglomerates, which can increase the tribological losses by accelerating the amount of three-body abrasion wear on the surface.

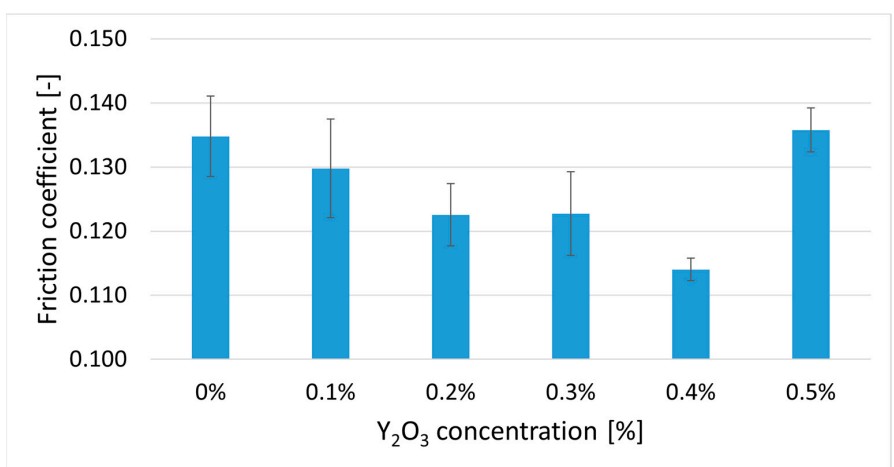

**Figure 12.** Measured friction coefficient values caused by the nanolubricants with different yttrium oxide concentrations.

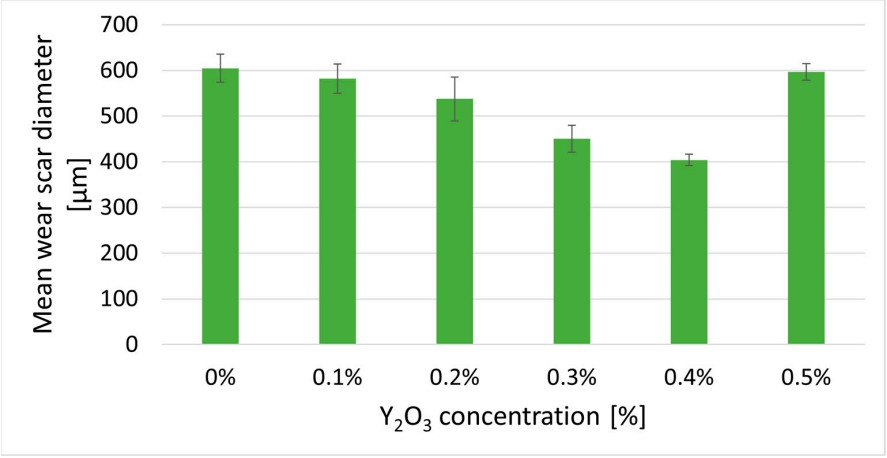

**Figure 13.** Measured mean wear scar diameter values caused by the nanolubricants with different yttrium oxide concentrations.

Figure 14 shows the comparison of digital microscope images captured of the worn surfaces of 0 and 0.4 wt% yttria-added samples. These images confirm the previously mentioned extremely positive anti-wear properties of the investigated $Y_2O_3$ nanoparticles: low dimensions, a low wear depth, visible original surface grooves and the main wear mechanisms of abrasion and polishing. Rainbow-colored spots can be seen in the wear marks on both the ball and the disc.

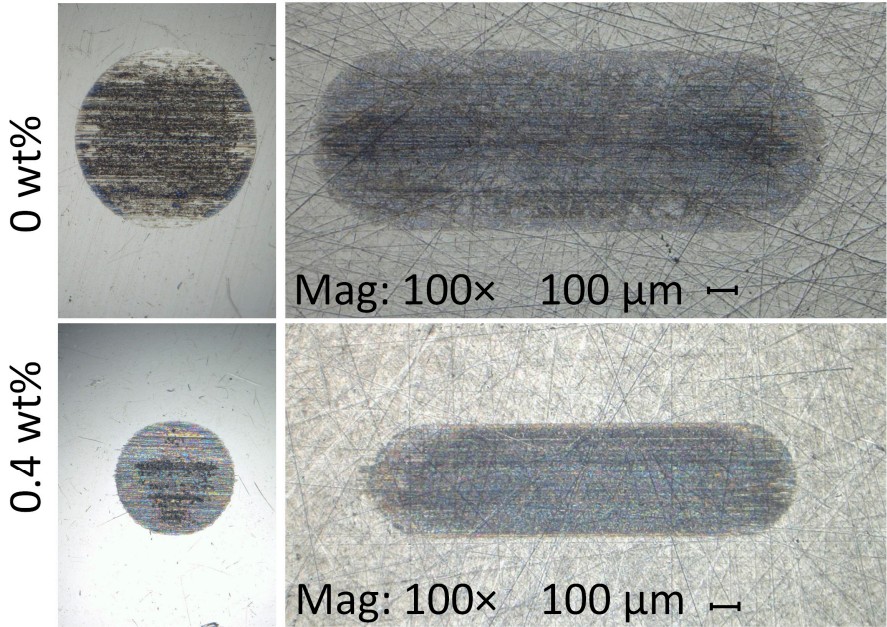

**Figure 14.** Digital microscope images of the worn surfaces of the ball (**left**) and disc (**right**) specimens with a 0 (**top**) and 0.4 wt% (**bottom**) yttrium oxide concentration nanolubricant sample.

During the analysis of the defined wear volume data (Figure 15), the excellent anti-wear ability of the yttria nanoparticles could be proved. Each yttria concentration decreased the measurable wear volume by at least 37% compared with the result of the reference sample without additional nanoparticles. The lowest definable wear volume was observed at a 0.4 wt% concentration with the wear decreasing potential of 77%. The tendency is identical to the defined tendency at the friction coefficient and the mean wear scar diameter: a continuous decrease until 0.4 wt% and an increase at 0.5 wt%.

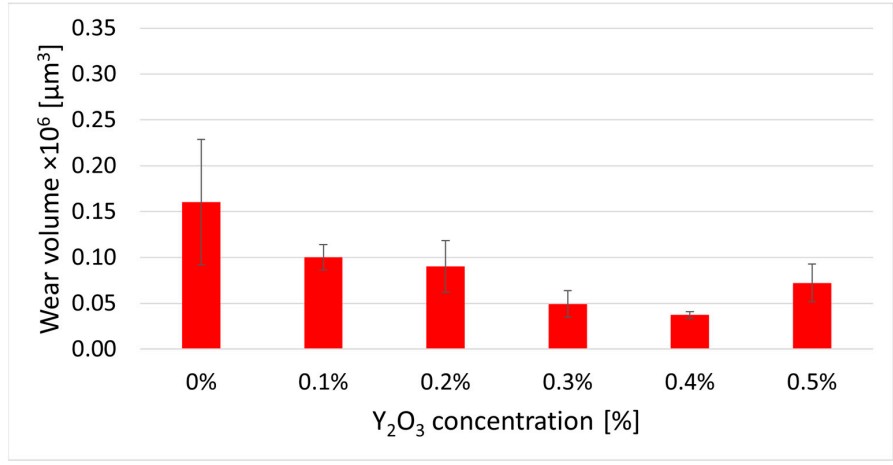

**Figure 15.** Measured wear volume values caused by the nanolubricants with different yttrium oxide concentrations.

Figure 16 shows the scanning electron microscopy images of the wear track of the disc tested with a 0.4 wt% yttria nanoadditive (left column), supplemented with the Y locations measured with EDX (right column). The figures in the upper row were taken in the middle point of the wear track, while the lower ones are in the dead center of the ball. Based on the electron microscopy images, it can be established that the wear is extremely low and the original surface structure barely changed. Few abrasion marks are visible in the direction of ball movement as the main type of wear. It can be observed that the worn surface is almost completely—except for the deep valleys—covered with dark spots from the protective boundary layer.

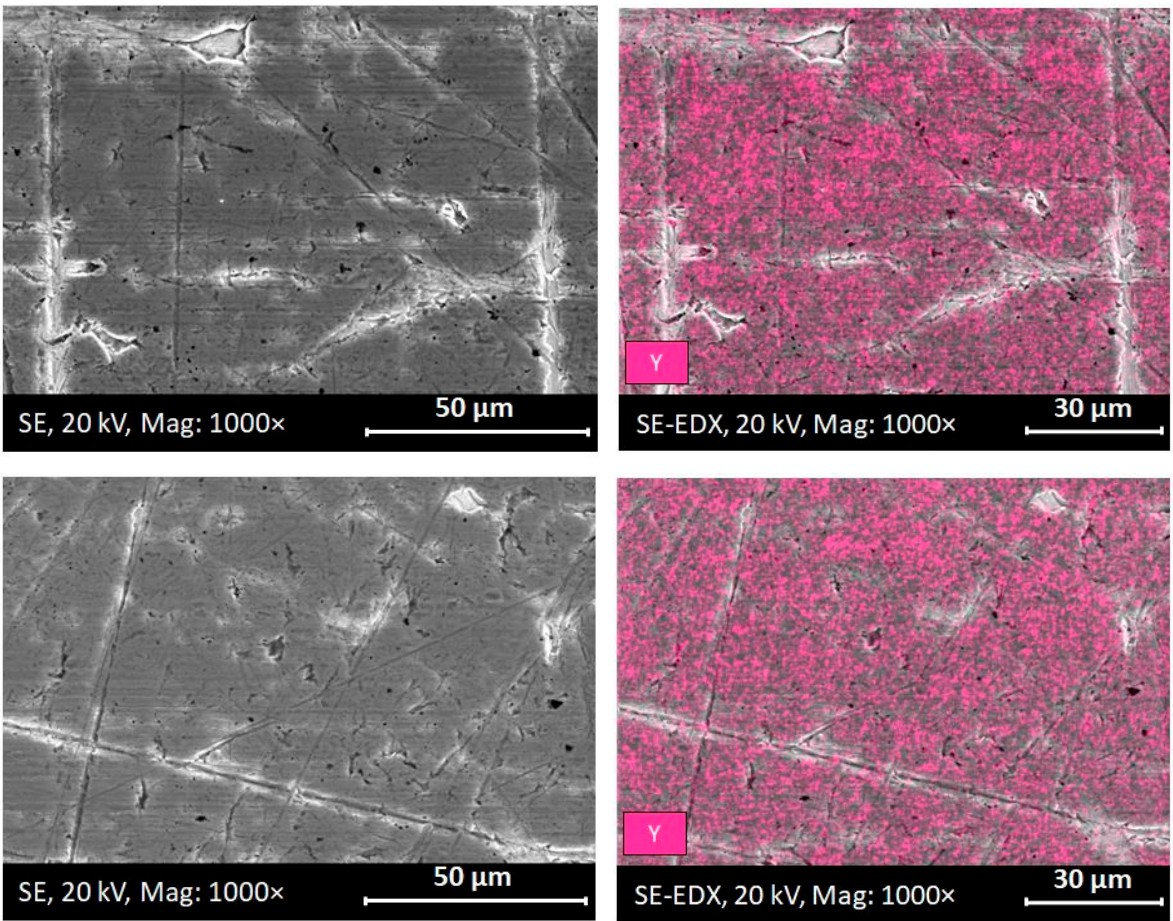

**Figure 16.** SEM and EDX images of the worn surface of the disc specimens measured with 0.4 wt% yttrium-oxide-added samples from its stroke-middle section (**top**) and dead center (**bottom**) areas with the horizontal sliding direction.

Table 3 shows the elemental quantification table of measurements performed with 0.4 wt% yttria nanoparticles. The presence of a large amount of yttrium can be observed on the Y map based on the results of the EDX analysis (see Figure 16), but on the other hand, with the Cu and Ti maps, yttrium does not occur equally in all areas. Only in those areas where the OBCS was able to form a protective layer (dark), yttrium is not found in the deeper parts of the surface. There is a large amount of yttrium on the surface: around the dead center, Y = 5.66 norm.wt%; in the middle of the wear mark, Y = 6.17 norm.wt%. A significant difference compared to the measurements made with CuO and TiO$_2$ nanoadditives is that there is much more calcium on the surface when the yttria nanoadditive is used. However, the trend of calcium occurrence is similar; it is higher in the dead center area (Ca = 1.88 norm.wt%) than in the middle area (Ca = 1.8 norm.wt%).

**Table 3.** Results of the SEM-EDX analysis from the investigated two different areas on the disc specimen measured with $Y_2O_3$-added lubricant samples; the result numbers are presented in normalized mass percent.

| Element | Stroke-Middle Section | Dead Center |
|---|---|---|
| Fe | 79.60 | 79.78 |
| Cr | 1.61 | 1.59 |
| Si | 0.41 | 0.35 |
| O | 7.84 | 8.18 |
| C | 2.50 | 2.49 |
| Ca | 1.80 | 1.88 |
| S | 0.07 | 0.07 |
| Y | 6.17 | 5.66 |

In summary, the surface-modified yttria nanoadditive with ethyl oleate works properly when used in Group III base oil with a 1 wt% OBCS content. When a yttria nanoadditive is used, the concentration of 0.4 wt% provides the best tribological properties in all respects: a 15% reduction in friction, 33% reduction in mean wear scar diameter and 77% reduction in wear volume. The wear rate is very low and typically abrasive. Based on the results, it can be concluded that nanoscale yttria and OBCS work well together, and both are excellently able to exert their effects. A strong synergistic effect can be observed regarding the coefficient of friction between the two additives. Compared to the author's previous work, the addition of yttria significantly reduced friction in the presence of OBCS [42]. During the interaction of the two additives, a boundary layer rich in calcium and yttrium is formed, which has strong anti-wear properties. In the future, it would be advisable to research the thin layer analysis of this boundary layer to better understand the operation of the yttria additive.

## 4. Discussion and Conclusions

Most studies either only look at the effects of individual additives and therefore use some simple base oil, or they use fully formulated lubricants. The novelty of this study is that it examines nanoparticles in a section of engine lubricant development. In other words, it examines compatibility with a real engine lubricant additive and the joint effect to make nanoparticles suitable for integration in the future. The advantage of this method is that a motor oil containing nanoparticles can be built up bottom-to-top. In this way, in the future, it will be possible to explore how nanoparticles can be effectively integrated and formulated into commercial engine lubricants.

The study experimentally demonstrated the tribological effect of three different ethyl oleate surface-modified transition metal oxide (copper(II) oxide, titanium dioxide and yttrium(III) oxide) nanoparticles in the presence of 1 wt% overbased calcium sulfonate. A linear oscillating tribometer was used for the tests, and microscopic methods were used to evaluate the worn samples. All three transition metal oxide nanoparticles were tested at concentrations of 0.1, 0.2... 0.5 wt% with four measurements each. During the evaluation of the results, the coefficient of friction, the mean wear scar diameter of the ball specimen and the volume of the wear of the disc specimen were determined. For a further analysis, scanning electron microscopic images of the worn surface were taken to determine the type of wear. Energy dispersive X-ray spectroscopy was used to determine the elemental composition of the tribofilm and the presence of nanoadditives. The main results are listed in the following points and summarized in Table 4:

- It was found that the tested oxide nanoparticles surface-modified with ethyl oleate can be effectively used in the presence of overbased calcium sulfonate.
- In the given tribology system, the optimal concentration of copper(II) oxide is 0.4 wt%, which resulted in a 4% reduction in friction, a 27% reduction in wear diameter and a 48% reduction in wear volume. The SEM+EDX analysis showed that copper(II) oxide

and the overbased calcium sulfonate together created a boundary layer with favorable tribological properties on the surface, with a copper content of 0.37 norm.wt%.

- In the tested tribology system, the optimal concentration of titanium dioxide is 0.4 wt%, which resulted in a 4% reduction in friction, a 24% reduction in wear diameter and a 63% reduction in wear volume. The SEM+EDX analysis showed that titanium dioxide and the overbased calcium sulfonate together created a boundary layer with favorable tribological properties on the surface, with a copper content of around 0.42 norm.wt%.
- In the given tribology system, the optimal concentration of yttrium(III) oxide is 0.4 wt%, which resulted in a 15% reduction in friction, a 33% reduction in wear diameter and a 77% reduction in wear volume. The SEM+EDX analysis showed that yttrium(III) oxide and the overbased calcium sulfonate have a synergic effect on each other. The two additives formed a strong boundary layer with favorable tribological properties on the surface, with a high calcium (1.8–1.88 norm.wt%) and yttrium content (5.66–6.17 norm.wt%).
- During the EDX analysis, it was established that the protective layer created with overbased calcium sulfonate has a higher calcium content in the dead ends of the wear track and a lower calcium content in the center of the wear track.

The results can be compared with the literature in a limited way because there has not been a nanoparticle additive test in the presence of OBCS. In summary, it can be concluded that the optimal concentration of all three nanoparticles is 0.4 wt%, which roughly corresponds to the expected value based on the literature. Based on the results of the literature and preliminary measurements, the friction-reducing effect of CuO and $TiO_2$ was somewhat lower than expected. In the case of both additives, despite the different concentrations, no or only a minimal reduction was observed. The examination of this requires a further analysis in the context of nanoparticles and OBCS. While their anti-wear effect, even in the presence of OBCS, agrees with the results expected based on the literature, the results of the measurements with the $Y_2O_3$ nanoparticle cannot be compared with previous literature results, but they worked using the preliminary tests. According to the literature, the boundary layer created with OBCS was also formed in the case of the present measurements, but the nanoparticles were able to reinforce it and make it more wear resistant.

**Table 4.** Summary of the tribological results in the optimum concentration (0.4 wt%) of each oxide nanoadditive. The table shows the deviation from the reference as a percentage.

| Result | 0.4% CuO | 0.4% $TiO_2$ | 0.4% $Y_2O_3$ |
|---|---|---|---|
| Coefficient of friction | −4% | −4% | −15% |
| Mean wear scar diameter | −27% | −24% | −33% |
| Wear volume | −48% | −63% | −77% |

In addition to the positive tribological effects, the results showed the good cooperation ability of the two groups of additives. Oxide ceramic nanoparticles appear promising for engine lubricant formulation studies. Testing with additional groups of additives is needed to ensure that they are suitable for functioning as additives in real motor oils in the future. The results reported in the article bring the applied oxide nanoparticles one step closer to becoming an additive in commercial engine lubricants in the future. There are still many questions surrounding the functioning of nanoparticles, so a thin-layer analysis of worn surfaces can help in this understanding.

**Author Contributions:** Conceptualization, Á.I.S.; methodology, Á.D.T. and Á.I.S.; formal analysis, Á.D.T. and Á.I.S.; investigation, Á.I.S.; resources, H.H.; data curation, Á.D.T. and Á.I.S.; writing—original draft preparation, Á.D.T. and Á.I.S.; writing—review and editing, Á.D.T. and Á.I.S.; visualization, Á.D.T. and Á.I.S.; supervision, H.H.; project administration, Á.D.T.; funding acquisition, Á.D.T. All authors have read and agreed to the published version of the manuscript.

**Funding:** This research received no external funding.

**Data Availability Statement:** Not applicable.

**Acknowledgments:** This article is published in the framework of the project "Synthetic fuels production and validation in cooperation between industry and university", project number: "ÉZFF/956/2022-ITM_SZERZ". The authors would like to thank Márk Marsicki, Achillesz Morvai and Jan Rohde-Brandenburger for their general support.

**Conflicts of Interest:** The authors declare no conflict of interest.

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
