# Peer review of "Tribological Investigation of the Effect of Nanosized Transition Metal Oxides on a Base Oil Containing Overbased Calcium Sulfonate"

_lubricants, doi:10.3390/lubricants11080337_

Round 1

Reviewer 1 Report

The manuscript has a thorough investigation and is therefore worthy of publication. However, the following issues have to be addressed before the final acceptance;

The authors should add a sentence of research background to the abstract. Demonstrate in the abstract novelty, practical significance.

After analyzing the literature, show before formulating the goal of the "blank" spots. Which has not been previously done by other researchers? You must show the importance of the research being undertaken. Show what will be the new research approach in this article. You need to show a hypothesis. In the last paragraph of the introduction, add scientific novelty and practical relevance. Add a clear purpose to the article.

The authors should add SEM or TEM images and particle size and APS graphs of as-received nanoparticles (copper(II) oxide, titanium dioxide, yttrium(III) oxide)

Please clarify why authors select these oxide nanoparticles why not different oxide-based nanoparticles ?

Please show wear traces in all SEM figures.   

Improve the results and discussion and conclusion parts. The results and discussion section should be widened with more focusing point of the findings. And these sentences should be supported with the literature studies. Results and discussion and conclusion parts are inadequate according to citation and analyze in detail. There should be the importance of the study in detail, comparison results with other approaches in literature, the success of the prediction and computational results.

In conclusion section, it is necessary to more clearly show the novelty of the article and the advantages of the proposed method. Add qualitative and quantitative results of your work. Please try to emphasize your novelty, put some quantifications, and comment on the limitations. This is a very common way to write conclusions for a learned academic journal. The conclusions should highlight the novelty and advance in understanding presented in the work.

Please fix the typographical and eventual language problems in paper.

Language used in the manuscript is generally satisfying. However, writers should pay more attention of singular / plural nouns. Also, they should control the spell check/ punctuation of words and sentences. Please check all manuscript for language and misspellings. Also, please recheck upper and lower case letter. . In addition, spaces should be added between words and numbers. Please fix the typographical and eventual language problems in paper.

Author Response

The authors would like to thank you for the feedbacks we have received. The answers and detailed modifications can be seen in the attached file.

Reviewer 2 Report

This manuscript reports a piece of interesting work on the effectiveness of three nanosized transition metal oxides on the tribological behaviour under oil lubricated conditions. The experiments were implemented systematically and the results are presented logically. But there are some issues that should be addressed in order to improve the quality of the paper, as follows.

1. Since CuO and TiO2 particle additions have been extensively studied, and Y2O3 has been studied by the authors recently, what is the novelty of this work? Please highlight this clearly in the Introduction and Conclusions.

2. Figure 1 showing the equipment is unnecessary since the facilities are described in Section 2.

3. Materials and Methods: After friction tests, how were the samples cleaned before microscopic examination? Please specify.

4. Results: Please compare the COF of CuO and TiO2 measured in this work with those reported in the literature.

5. Overall, the paper lacks comparison with published work. Very few references were cited in the Results section. Only two own references [18], [39] were used.

6. Scientific interpretation: Authors need to explain the following phenomena with experimental evidences:

(1) Why 0.2% Cu and 0.2% TiO2 caused increased friction and/or wear volume? Please provide evidence to show the difference in wear morphology between different % of particles. In the current version, only the wear morphology with 0.4% particles is provided.

(2) Similarly, why 0.5% Y2O3 caused and increase in COF and ball wear?

7. Please indicate 0.4% particle in the captions of Figure 6, 11, and 16.

8. Conclusions: The first paragraph should be rewritten to highlight the novelty and significance of the work.

9. Conclusions: From the results presented, it is clear that Y2O3 is the most effective among the three oxides studied. This should be highlighted in the Conclusions and discussed in Section 3. May consider using a table to compare three oxides under respective optimum conditions.

It is generally fine.

Author Response

(The authors gave the same response as above.)

Reviewer 3 Report

The manuscript presents an investigation into the degradation behavior of methylammonium lead bromide (MAPbBr3) single crystals under simultaneous oxygen and light illumination in ultrahigh vacuum (UHV) conditions. The degradation process was monitored using X-ray photoelectron spectroscopy (XPS), allowing precise control over exposure time and oxygen pressure. The results indicate that the combined exposure to oxygen and light accelerated the degradation of MAPbBr3, suggesting that it cannot be attributed solely to the additive effects of oxygen or light alone. The XPS spectra revealed a significant loss of carbon, bromine, and nitrogen when exposed to an oxygen pressure of 10^10 Langmuir with light illumination. The study was conducted meticulously, but there are some aspects that require further clarification. Here are my detailed comments and questions:

1.     The inclusion of Figure 1, showing photographs of the equipment, seems unnecessary unless the equipment possesses unique features or is home-built. Therefore, I suggest removing Figure 1.

2.     It would be helpful if the authors could provide information on the sizes of the copper(II) oxide, titanium dioxide, and yttrium(III) oxide nanoparticles. Additionally, it would be beneficial to include microscopic images such as high-resolution SEM or TEM images to demonstrate the detailed microscopic structures of the nanoparticles and provide information on the size distribution of the nanoparticles.

3.     The manuscript mentions that oxide nanoparticles can reduce friction, and the introduction of nanoparticles for lubrication purposes is an important strategy. However, there are several crucial factors to consider, such as morphology and ligands. It would be relevant to mention and cite earlier works that have investigated these aspects at the nanometer scale, such as the studies conducted by V. S. Jatti et al. (Journal of Mechanical Science and Technology 29, 793, 2015), Y. Yuk et al. (Nanotechnology 26, 135707, 2015), M. Waqas et al. (Materials 14, 6310, 2021), and J. Y. Park (Langmuir 27, 2509-2513, 2011).

Overall, the experimental results are intriguing, and the study demonstrates meticulous execution. However, addressing these points would provide further clarity and enhance the relevance of the research by incorporating previous investigations in the field.

Moderate improvement of English is required. 

Author Response

(The authors gave the same response as above.)

Round 2

Reviewer 3 Report

I suggest the paper be accepted. 

I suggest the paper be accepted.